# Knowledge, Attitudes and Practices of the General Population in Yemen Regarding COVID-19: A Cross-Sectional Study

**DOI:** 10.3390/diseases11010017

**Published:** 2023-01-26

**Authors:** Mohammed Saif Anaam

**Affiliations:** 1Department of Pharmacy Practice, Unaizah College of Pharmacy, Qassim University, Buraydah 999088, Saudi Arabia; m.anaam@qu.edu.sa; 2Al-Rowaad Medical College, Sana’a 5696, Yemen

**Keywords:** COVID-19, attitude, knowledge, practice, Yemen

## Abstract

Background: Following the World Health Organization declaration of COVID-19 as a pandemic, Yemen has taken preventive and precautionary measures against COVID-19 to control its spread. This study evaluated the knowledge, attitudes, and practices (KAP) of the Yemeni public regarding COVID-19. Methods: A cross-sectional study using an online survey was conducted during the period from September 2021 to October 2021. Results: The mean total knowledge score was 9.50 ± 2.12. The majority of the participants (93.4%) were aware that to prevent infection by the COVID-19 virus, going to crowded places and gatherings should be avoided. Approximately two thirds of the participants (69.4%) believed COVID-19 is a health threat to their community. However, in terms of actual behavior, only 23.1% of the participants reported that they did not go to any crowded places during the pandemic, and only 23.8% had worn a mask in recent days. Moreover, only about half (49.9%) reported that they were following the strategies recommended by the authorities to prevent the spread of the virus. Conclusion: The findings suggest that the general public has good knowledge and positive attitudes regarding COVID-19, but that their practices are poor.

## 1. Introduction

Coronavirus disease 2019 (COVID-19) is an infectious disease caused by a novel coronavirus called Severe Acute Respiratory Syndrome Coronavirus 2 (SARS-CoV-2) [1]. The virus was first discovered in Wuhan city, Hubei Province, China and reported to the World Health Organization (WHO) in December 2019 [2,3]. COVID-19 has spread widely and rapidly, from Wuhan city, to other parts of the world, threatening the lives of many people. The main symptoms of COVID-19 have been identified as fever, dry cough, fatigue, myalgia, and shortness of breath or dyspnea [4,5]. In severe cases, it can cause pneumonia, respiratory failure, cardiac arrest and death [6]. Nevertheless, some studies reported estimates that from about 30% to 70% of patients could contract the virus without showing symptoms of the disease [7,8,9,10,11,12], impacting the control of the pandemic. Standard preventive procedures including social distancing, regular hand sanitization, regular wiping of surfaces, quarantining and wearing of face masks have been adopted worldwide as the most effective methods of reducing the spread of the virus and limiting its morbidity and mortality [13,14,15,16,17]. While many treatments have been suggested to treat patients with COVID-19, only a few pharmacological therapies have been shown to be beneficial in the management of COVID-19 [18,19,20,21,22]. Therefore, many governments and health authorities, including the WHO, have been actively educating people to take preventive measures to decrease the spread of the virus [23,24,25,26]. By the end of January 2020, the World Health Organization (WHO) declared COVID-19 as a pandemic and announced a public health emergency of international concern requiring collaborative efforts among all countries to prevent its rapid spread [27]. Following the WHO declaration, countries around the globe, including Yemen, have adhered to global plans aimed at bringing the pandemic under control. The first confirmed case of COVID-19 was reported in Yemen on 10 April 2020 [28]; following this, the Yemeni government introduced proactive and early precautionary measures to help reduce the spread of COVID-19. These proactive measures included travel restrictions, closing commercial centers and enforcing lockdown measures, implementing curfews, suspending school academic activities (including those of universities) and prayers at mosques, and limiting the number of employees at workplaces. Nevertheless, due to the unstable political situation, strong preventive and precautionary measures were very difficult to implement. The scenario of COVID-19 in Yemen was obscure, and the pandemic spread appeared rather mild, which may be real or attributable to possible underreporting of cases. However, from 10 April 2020 to 20 November 2021, there were 9977 confirmed cases of COVID-19 with 1943 deaths, reported to WHO [29]. Despite the fragile and unstable political situation in the country, public adherence to preventive measures established by the government is of prime importance in preventing the spread of the disease. Public adherence is likely to be influenced by knowledge of and attitudes toward COVID-19. Evidence suggests the importance of public knowledge in tackling pandemics [30,31]. Assessing the public’s knowledge about the coronavirus may provide deeper insights into existing public perceptions and practices, which would help to identify attributes that influence the public in adopting good practices towards COVID-19 [32]. Assessing public knowledge is also important in identifying gaps and strengthening ongoing efforts to control the pandemic. Therefore, this study aimed to investigate the knowledge, attitudes and practices of the general population in Yemen with regard to COVID-19.

## 2. Methodology

### 2.1. Study Design and Location

A cross-sectional study using an online survey was conducted. The online survey was administered using Google Forms. The study took place over a period of 2 months, from September 2021 to October 2021, in Yemen.

### 2.2. Study Population, Inclusion and Exclusion Criteria

The targeted population in this study was the general population. Individuals willing to participate in the study and aged ≥15 years were included in the study. Those not willing to participate and aged <15 years were excluded.

### 2.3. Sample Size Determination and Sampling Technique and Questionnaire Administration

Raosoft software [33] was used for sample size calculation, considering 95% as a confidence level, 5% as a margin of error and 50 % as a response distribution. Consequently, the required minimum sample size was estimated to be 377 participants in this study.

Through a convenience sampling method, invitations were sent to potential participants through WhatsApp, WhatsApp groups, and the Facebook platform including a link to the online survey. The invitation message included a brief explanation about the study and its objectives and emphasized confidentiality and the use of personal data for the scientific work only. Moreover, several reminders to encourage potential responders to complete the survey were sent. In administering the survey, duplicate responses were avoided by limiting the response from the same device to only one response. Consent to take part in the study was assumed once the participants responded to the online survey. In this research, participation was completely voluntary, and no rewards were provided.

### 2.4. Development of the Questionnaire

A questionnaire for the investigation was developed on the basis of a previously published study [34] with permission from the main author, but the scale was adapted to the research context through minor changes in the wording of the items. The overall reliability of the KAP questionnaire was observed to be 0.714, which was found to be above the recommended value of 0.7 [35,36]. The final draft instrument was developed after an expert discussion with three academicians on the questionnaire design and was slightly modified. The modifications were minor and linguistic in nature to be suitable in the local context of Yemen. In addition, the questionnaire survey was sent to 15 individuals for review to ensure appropriate length, structure, and comprehensiveness of the instrument; their feedback confirmed that the survey was clear and easy to understand. The data of the pilot study were excluded from the final analysis. The final questionnaire contained 12 statements to examine respondents’ knowledge about COVID-19. A correct response was given a score of 1, and an incorrect or “unsure” response was given a score of 0. The maximum knowledge score was 12. There were three attitude questions and three practice questions as well as eight questions related to demographic data. The six questions on attitude and practice were framed using three possible answers (“Yes”, “No”, “Not sure”). A score of 1 was awarded for each positive reaction toward practice questions, and a score of zero for a negative reaction. “Not sure” was recorded as a negative reaction. The maximum practice score was 3. 

### 2.5. Data Analysis

The data were analyzed using Statistical Package for Social Sciences software (SPSS, version 21, SPSS, Chicago, IL, USA). Descriptive statistics (i.e., frequencies, percentages, means, and standard deviations) were used to present the responses of the participants to the KAP statements. Inferential statistics were used to establish the association between demographics and KAP level. Pearson’s Chi-square or Fisher’s exact test was used to examine the association between categorical variables. Kolmogorov–Smirnov (K–S test) was used to test for the normal distribution of the continuous variables before performing inferential statistical tests. Independent t-test and one-way ANOVA were used to examine the difference in means of the continuous variables between the targeted groups. A *p*-value of <0.05 was considered statistically significant.

### 2.6. Ethics Statement

The study was approved by the ethical Committee at Alrowad Medical college (protocol code A-21-47 and approved on 12 July 2021). The study was conducted according to the Declaration of Helsinki.

## 3. Results

### 3.1. Demographic Characteristics 

A total of 467 respondents participated in the current study. The mean age of the participants was 30.1 ± 10.7 years, and the majority of participants were males (69.4%). Most of the participants (*n* = 282; 60.4%) were in the younger age group (15–29 years). Slightly more than half (52.5%) of the respondents were single. In this study, 16.7% of the participants reported that they had COVID-19, and 22.9% reported that a member of their family had COVID-19. The vast majority of participants (*n* = 415; 88.9%) did not have any coexisting diseases. Demographic characteristics of the participants are shown in Table 1.

### 3.2. Knowledge about COVID-19 

The mean total knowledge score was 9.50 ± 2.12 out of the maximum score of 12. The overall rate of correct answers for the knowledge statements was 79.2%. Table 2 shows responses to the knowledge questions in detail. T-test and one-way ANOVA were performed to examine whether there were any statistically significant differences in means of the participants’ knowledge score in terms of their demographic characteristics. Statistically significant differences (*p* < 0.05) in means of knowledge scores were found among genders, level of education, and those who had COVID-19 (Table 3). The female participants had a higher level of knowledge (9.77 ± 1.81) compared to the male participants (9.38 ± 2.23) in this study (*p* = 0.047). 

### 3.3. Attitudes toward COVID-19

In this study, 70.1% of study population positively answered the attitude questions. More than half of the participants (61.5%) agreed that COVID-19 will finally be successfully controlled. The majority (69.4%) of the participants believed that COVID-19 is a health threat to the community. Moreover, more than two thirds of participants (79.4%) were of the opinion that lockdowns would improve the situation in Yemen. Table 4 summarizes participants’ responses to the attitude questions. Chi-square analysis showed that there were statistically significant associations between attitude in terms of agreement that COVID-19 will finally be successfully controlled and gender (*p* = 0.041), those who had a family member ever infected with coronavirus (*p* = 0.037), and those with coexisting disease (*p* = 0.007). Meanwhile, the attitude in terms of thinking that COVID-19 is a threat to the community was statistically significantly associated with occupation (*p* = 0.046) and marital status (*p* = 0.047). T-test showed a statistically significant association between knowledge and attitude (*p* < 0.001) through all attitude questions (Table 5).

### 3.4. Practices toward COVID-19 

The total practice score was 3, while the mean score was 0.97 ± 0.96. Study results showed that 331 participants (70.9%) had poor practices. Only 108 participants (23.1%) did not go to any crowded places, and 111 participants (23.8%) reported wearing a mask when leaving home. Moreover, about half of participants (*n* = 233; 49.9%) only reported that they were following the strategies recommended by the authorities to prevent infection and spread of COVID-19. The details of the replies to the practice questions are summarized in Table 6. Chi-square analysis showed that there was a statistically significant association between the practice of going out to crowded places during the pandemic and gender (X^2^ = 5.6; *p* = 0.018), education (X^2^ = 11.1; *p* = 0.004), and occupation (X^2^ = 15.1; *p* = 0.004). One-way ANOVA and t-test showed statistically significant differences in means of the participants’ practice scores among occupation (F = 2.6; *p* = 0.039) and level of education categories (t = 2.1; *p* = 0.037), respectively. 

## 4. Discussion

The present study examined the knowledge, attitudes and practices with regard to COVID-19 in 467 participants who took part in this survey. This is probably the first study in Yemen. The general population appeared to have good knowledge about COVID-19, with an overall rate of 79.2%. The vast majority of respondents gave correct answers for most knowledge statements, which ranged between 45.4–93.4%. The findings of this study are similar to previous studies conducted in other countries such as Sudan (78.2%) [37], Jordan (80.0%) [38], Malaysia (80.5%) [39], and India (81.0%) [40]. The Yemeni population’s level of knowledge is higher than that reported in other studies such as in Egypt (70.2%) [41], Bangladesh (69.8%) [42], Nigeria (65.4%) [43] and Liberia (51.0%) [44], but it is lower than that reported in some other countries such as Cameroon (84.2%) [45], Bangladesh (85.0%) [46], Saudi Arabia (89.9%) [47], China (91.2%) [48], and Uganda (93.9%) [49]. In this study, one area of concern was the fact that only 45.4% of participants correctly answered the statement, “eating or contacting wild animals would result in the infection by the COVID-19 virus”, while 39.2% were not sure. However, the participants in our study had better knowledge compared to the findings from Malaysian [39], Bangladesh [50] and Ethiopian studies [51], where only 35.7%, 39.8% and 10.6% of participants respectively stated that transmission from animals is likely. A worrying finding was that even among individuals with good knowledge (*n* = 286), high percentages still had poor practices. In response to the first, second and third practice questions, the percentage of poor practices among those individuals were 76.6%; 74.5%, and 47.6%, respectively.

Regarding attitudes, the majority of our participants believed COVID-19 is a serious disease and poses a health threat to the community, with 79.4% believing that lockdowns and curfews during the early stages of the pandemic were required for controlling the pandemic but may harm society’s economic situation. This is similar to previous studies conducted in Saudi Arabia [52,53,54], India [40,55], and Pakistan [56]. In line with previous studies [40,48,50,55,57], Yemenis expressed an optimistic attitude towards the COVID-19 pandemic, where more than half (61.5%) of the study participants believed that the pandemic would be successfully controlled. Although a majority of study participants believed COVID-19 is a serious disease, and they were also aware that the infection can be prevented by avoiding crowded places and gatherings, a majority (71.3%) still attended crowded places and about two thirds (73.2%) did not wear a mask when leaving home. Moreover, only about half (49.9%) of the participants reported that they followed the strategies recommended by authorities to prevent infection with and spread of COVID-19. These potentially risky behaviors were related to gender, education and occupation. As suggested by findings from previous studies regarding age and gender patterns of risk-taking behaviors [58,59], men and late adolescents were more likely to engage in risk-taking behaviors. In line with these previous findings, we found a significant association between male gender and potentially dangerous practices regarding COVID-19 in this study. Among males, the significantly higher risk of going to a crowded place could be ascribed to their desire to gather with others to spend time, discuss daily issues and chew Khat (*Catha edulis*, whose leaves are chewed for their stimulating effect), a social habit widespread throughout the country [60]. Among the general population, the significantly higher risk of not wearing a mask when leaving home could be due to their misconceptions about the importance of masks and preventive measures. However, this poor practice also could be due to the absence of strong precautionary directives or penalties enforced by the government. Although there was no statistically significant association between knowledge and practice, we noticed that the likelihood of potentially dangerous practices regarding the COVID-19 pandemic was not minimized by having good knowledge about the pandemic.

## 5. Strengths and Limitations

This study is the first of its kind in Yemen that addressed the knowledge, attitudes and practices of the general population regarding COVID-19. Nevertheless, the survey method using the internet may bias the sample towards participants from urban areas who have better access to the internet. Additionally, one of the limitations is the bias of online surveys towards younger generations which is well documented in the literature. Further, self-reporting surveys may cause social desirability bias, where participants may have answered attitude and practice questions positively based on what they perceive to be expected from them.

## 6. Conclusions

The findings suggested that the general public in Yemen has good knowledge and positive attitudes regarding COVID-19. However, the study findings highlighted poor practices regarding preventive measures, including wearing masks and social distancing. Health education campaigns and awareness events targeting the general population could be helpful to improve practices in facing the crisis. In addition, further studies are needed to address differences in attitudes and practices in specifically vulnerable populations, as well as about people’s KAP vis-à-vis SARS-CoV-2 vaccines. 

## Figures and Tables

**Table 1 diseases-11-00017-t001:** Demographic Characteristics (*n* = 467).

Characteristic	*n* (%)
**Gender**	
Male	324 (69.4)
Female	143 (30.6)
**Age groups**	
15–29	282 (60.4)
30–44	130 (27.8)
45–59	38 (8.1)
≥60	17 (3.6)
**Occupation**	
Employed	205 (43.9)
Student	154 (33.0)
Housewife	26 (5.6)
Unemployed	32 (6.9)
Others (self-employed)	50 (10.7)
**Marital status**	
Single	245 (52.5)
Married	209 (44.8)
Divorced	8 (1.7)
Widowed	5 (1.1)
**Level of education**	
School	66 (14.1)
University	401 (85.9)
**Had COVID-19**	
Yes	78 (16.7)
No	389 (83.3)
**Had family member with COVID-19**	
Yes	107 (22.9)
No	360 (77.1)
**Coexisting disease**	
Yes	52 (11.1)
No	415 (88.9)

**Table 2 diseases-11-00017-t002:** Responses to Knowledge Questions (*n* = 467).

Item	Correct *n* (%)	Not Correct *n* (%)	Don’t Know *n* (%)
K1. The main clinical symptoms of COVID-19 are fever, fatigue, dry cough, and muscle pain.	393 (84.2)	24 (5.1)	50 (10.7)
K2. Unlike the common cold, stuffy nose, runny nose, and sneezing are less common in persons infected with the COVID-19 virus.	289 (61.9)	73 (15.6)	105 (22.5)
K3. Currently there is no effective cure for COVID-19, but early symptomatic and supportive treatment can help most patients to recover from the infection.	380 (81.4)	24 (5.1)	63 (13.5)
K4. Not all persons with COVID-19 will develop severe cases. Those who are elderly, have chronic illnesses, and are obese are more likely to be severe cases.	398 (85.2)	32 (6.9)	37 (7.9)
K5. Eating or contacting wild animals would result in the infection by the COVID-19 virus.	212 (45.4)	72 (15.4)	183 (39.2)
K6. Persons with COVID-19 cannot spread the virus to others when the symptoms of COVID-19 are not present.	279 (59.7)	91 (19.5)	97 (20.8)
K7. The COVID-19 virus spreads via respiratory droplets of infected individuals.	434 (92.9)	8 (1.7)	25 (5.4)
K8. Ordinary individuals can wear general medical masks to prevent the infection by the COVID-19 virus.	411 (88.0)	34 (7.3)	22 (4.7)
K9. It is not necessary for children and young adults to take measures to prevent the infection by the COVID-19 virus.	392 (83.9)	40 (8.6)	35 (7.5)
K10. To prevent the infection by COVID-19, individuals should avoid going to crowded places and avoid gatherings.	436 (93.4)	13 (2.8)	18 (3.9)
K11. Test, Trace and Isolate (TTI) are the effective ways to reduce the spread of COVID-19.	393 (84.2)	18 (3.9)	56 (12.0)
K12. People who have contact with someone infected with the COVID-19 virus should be immediately isolated in a proper place. In general, the observation period is 14 days.	419 (89.7)	24 (5.1)	24 (5.1)

**Table 3 diseases-11-00017-t003:** The difference in means of knowledge score among study participants (*n* = 467).

Variables	*n*	Knowledge	
		Mean (±SD)	*p*-Value *
**Gender**			0.047 *
Male	324	9.38 (2.23)	
Female	143	9.77 (1.81)	
**Age groups**			0.53
15–29	282	9.48 (2.02)	
30–44	130	9.44 (2.25)	
45–59	38	9.97 (1.90)	
≥ 60	17	9.29 (3.04)	
**Occupation**			0.099
Employed	205	9.61 (2.07)	
Student	154	9.44 (1.89)	
Housewife	26	9.81 (1.88)	
Unemployed	32	8.56 (2.99)	
Others	50	9.66 (2.32)	
**Marital status**			0.097
Single	254	9.46 (2.06)	
Married	209	9.62 (2.11)	
Divorced	8	7.75 (3.54)	
Widowed	5	9.20 (1.64)	
**Level of education**			0.022 *
School	66	8.85 (2.50)	
University	401	9.61 (2.03)	
**Had COVID-19**			0.011 *
Yes	78	10.1 (1.75)	
No	389	9.39 (2.17)	
**Had family member with COVID-19**			0.072
Yes	107	9.82 (2.05)	
No	360	9.40 (2.13)	
**Coexisting disease**			0.27
Yes	52	9.19 (2.41)	
No	415	9.54 (2.08)	

Note: Independent-samples *t*-test and one-way ANOVA were used; * Significant at *p* < 0.05.

**Table 4 diseases-11-00017-t004:** Attitudes toward COVID-19 among participants.

Questions	Yes *n* (%)	No *n* (%)	Don’t Know *n* (%)
A1. Do you agree that COVID-19 will finally be successfully controlled?	287 (61.5)	43 (9.2)	137 (29.3)
A2. Do you think that COVID-19 is a threat for your community?	324 (69.4)	95 (20.3)	48 (10.3)
A3. I think that the lockdown would improve the overall wellbeing of society in terms of controlling the COVID-19 pandemic situation.	371 (79.4)	46 (9.9)	50 (10.7)

**Table 5 diseases-11-00017-t005:** Association between knowledge and attitudes.

Variables	*n*	Knowledge	
		Mean (± SD)	*p*-Value *
A1. Do you agree that COVID-19 will finally be successfully controlled?			<0.001 *
Yes	287	9.79 (1.82)	
No/Unsure	180	9.04 (2.45)	
A2. Do you think that COVID-19 is a threat for your community?			<0.001 *
Yes	324	9.77 (1.83)	
No/Unsure	143	8.90 (2.56)	
A3. I think that the lockdown would improve the overall wellbeing of society in terms of controlling the COVID-19 pandemic situation.			<0.001 *
Yes	371	9.84 (1.70)	
No/Unsure	96	8.18 (2.92)	

Note: Independent-samples *t*-test was used. * Significant at *p* < 0.001.

**Table 6 diseases-11-00017-t006:** Practices toward COVID-19 among participants.

Questions	Yes *n* (%)	No *n* (%)	Don’t Know *n* (%)
P1. In recent days, have you gone to any crowded place?	333 (71.3)	108 (23.1)	26 (5.6)
P2. In recent days, have you worn a mask when leaving home?	111 (23.8)	342 (73.2)	14 (3.0)
P3. Are you following the strategies recommended by authorities (e.g., Ministry of Health) to prevent COVID-19 infection and spread?	233 (49.9)	177 (37.9)	57 (12.2)

## Data Availability

The datasets generated during and/or analyzed during the current study are available from the corresponding author on reasonable request.

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
