# Peer review of "Knowledge, Attitudes and Practices of the General Population in Yemen Regarding COVID-19: A Cross-Sectional Study"

_diseases, 2023, doi:10.3390/diseases11010017_

Round 1
Reviewer 1 Report
There is conflicting information about the dates for COVID-19's first appearance -- line 44 mentions April and January, and I assume it should be April. Lines 54-55 refer to a period 3 January 2020 to 20 November 2021 - but presumably the first case was only detected on 10 April, and the pandemic declared only by the second half of January.
In order to be considered a fully fledged scientific articles, the author should go much more into depth about the possible biases introduced by the methodology - this issue is rather superficially glossed over in section 5, even though the results reflect clear biases to certain sub-populations and age groups and probably also gender - this affects the links between KAP characteristics and other parameters. The list of references is rather too exhaustive for a study of this size and can be rationalized. It is a shame this survey was not extended to include people's KAP vis-à-vis SARS-CoV-2 vaccines. It is not clear whether information was collected that could link people's knowledge, attitudes and practices in relation to the wealth quintile into which they fit. The chewing of Khat is a social habit very specific to Yemen, and again the questionnaire could have explored the links with this social habit and attitudes/practice related to COVID-19 much more vigorously. Finally, can anything be said about the differences in attitude and practice in specifically vulnerable population groups? And suggestions for further research and for further actions by the health authorities.
Author Response
|
Comment |
Response |
|
Comments to the Author |
|
|
1. There is conflicting information about the dates for COVID-19's first appearance -- line 44 mentions April and January, and I assume it should be April. |
1. Thank you for bringing my attention to this. Recommendation was applied. Corrected it is in April it was typing error.
|
|
2. Lines 54-55 refer to a period 3 January 2020 to 20 November 2021 - but presumably the first case was only detected on 10 April, and the pandemic declared only by the second half of January.
|
2. Recommendation was applied. It is 10th April (typing error)
|
|
3. In order to be considered a fully-fledged scientific articles, the author should go much more into depth about the possible biases introduced by the methodology - this issue is rather superficially glossed over in section 5, even though the results reflect clear biases to certain sub-populations and age groups and probably also gender - this affects the links between KAP characteristics and other parameters.
|
3. The population who has more access and use internet is the young male people (especially) compared to female as the community is still conservative towards internet use by females who have also less access to social media in general, furthermore the old people don’t care about use internet in general. So the higher % of participants of young males is expected. This bias of online surveys towards younger generations is also documented in the literature.
|
|
4. The list of references is rather too exhaustive for a study of this size and can be rationalized. It is a shame this survey was not extended to include people's KAP vis-à-vis SARS-CoV-2 vaccines.
|
4. All references are related to the topic, and the main objective of the study was to evaluate the knowledge, attitudes, and practices (KAP) of the Yemeni public towards COVID-19 but not about vaccine so it is a good idea to study KAP about vaccines but the study has been already conducted and maybe we can conduct another study about vaccine in the future although there are few people took vaccine due to shortage of vaccine in the country because of the political instability. |
|
5. It is not clear whether information was collected that could link people's knowledge, attitudes and practices in relation to the wealth quintile into which they fit. |
5. There was no association between occupation and knowledge and this can be an indicator about the wealth quintile, unfortunately, we did not collect information about income. We adopt a questionnaire from the previous literature. |
|
6. The chewing of Khat is a social habit very specific to Yemen, and again the questionnaire could have explored the links with this social habit and attitudes/practice related to COVID-19 much more vigorously.
|
6. Unfortunately, we did not collect information about Khat chewing. As you mentioned, rightly, this social habit very specific to Yemen, that is why it is not included in the questionnaire which we had adopted. |
|
7. Finally, can anything be said about the differences in attitude and practice in specifically vulnerable population groups? And suggestions for further research and for further actions by the health authorities.
|
7. As the study completed it is difficult to get such information, but it is a good idea to be covered in future studies as mentioned in the text (Further studies are needed to address differences in attitude and practice in specifically vulnerable population groups as well as about people's KAP vis-à-vis SARS-CoV-2 vaccines). |
Reviewer 2 Report
This is a what's app survey to address the knowledge, attitude and practice of the population in Yemen towards COVID-19. In the section "strengths and Limitations" it should be discussed whether 467 participants with a mean age of the 30 years and 70 % male really are representative for the population of Yemen.
Author Response
|
Comment |
Response |
|
Comments to the Author |
|
|
1. This is a what's app survey to address the knowledge, attitude and practice of the population in Yemen towards COVID-19. In the section "strengths and Limitations" it should be discussed whether 467 participants with a mean age of the 30 years and 70 % male really are representative for the population of Yemen. |
Thank you
|
Reviewer 3 Report
The paper is well written, clear and easy to read.
Only one minor correction in line 201
Re lines 185, 186 on comparison with paper from other countries - 2 paper from Bangladesh (ref 42 and 46) were cited, one showing Yemen having better and the other worse performance compared to Bangladesh. Suggest to clarify the differences.
Regarding Strengths and Limitations (Line 227) suggest include "... in Yemen" when stating the strength of the present study.

Author Response
|
Comment |
Response |
|
Comments to the Author |
|
|
1. Re lines 185, 186 on comparison with paper from other countries - 2 paper from Bangladesh (ref 42 and 46) were cited, one showing Yemen having better and the other worse performance compared to Bangladesh. Suggest to clarify the differences.
|
Thank you for the comment
|
|
3. Regarding Strengths and Limitations (Line 227) suggest include "... in Yemen" when stating the strength of the present study.
|
3.Thank you for bringing my attention to this. Recommendation was applied.
|
Round 2
Reviewer 2 Report
Clearly improved manuscript